# Evaluation of a point-of-care rapid diagnostic test kit (SICKLECHECK) for screening of sickle cell diseases

**Prasanta Purohit**[1], **Chinmayee Parida**[1], **Tapan Kumar Martha**[1], **Snehal Bholo**[1], **Aparupa Naik**[1], **Samira Kumar Behera**[1,2]*

**1** Multi-Disciplinary Research Unit, Maharaja Krishna Chandra Gajapati (M.K.C.G) Medical College, Berhampur, Odisha, India, **2** Department of Pathology, Maharaja Krishna Chandra Gajapati (M.K.C.G) Medical College, Berhampur, Odisha, India

* samirbehera41@gmail.com

**Data Availability Statement:** All relevant data are within the manuscript and its Supporting Information files.

## Abstract

Sickle cell diseases (SCD) are the most common genetic disorders with significant morbidity and mortality worldwide, including in India. The high prevalence of this disorder in many geographical regions calls for the use of a point-of-care rapid diagnostic test (RDT) for early screening and management of the diagnosed cases to reduce the allied clinical severity. In view of this, the present study was undertaken for the validation of a point-of-care RDT kit (SICKLECHECK™) for the screening of SCD. This validation and diagnostic accuracy study was conducted among the cases advised for screening of SCD. For validation, all the recruited cases were investigated for both the SICKLECHECK™ RDT kit and HPLC (Variant-II) considering HPLC as a gold standard. A total of 400 cases were screened for both tests. For the presence and absence of sickle cell hemoglobin in the samples, SICKLECHECK™ RDT kit results showed a sensitivity and specificity of 99.39% and 98.73% respectively with references to HPLC findings. For the detection of the 'AS' pattern, the SICKLECHECK™ RDT kit has shown a sensitivity and specificity of 99.07% and 98.81% respectively. For the detection of the 'SS' pattern, the SICKLECHECK™ RDT kit has shown a sensitivity and specificity of 97.92% and 100.0% respectively. Cases with β thalassemia trait, hemoglobin E trait, hemoglobin Lepore trait and trait for hereditary-persistence-of-fetal-hemoglobin (high HbF %) diagnosed in HPLC were resulted with 'AA' pattern in SICKLECHECK™ RDT kit. The high sensitivity and specificity of the SICKLECHECK™ RDT kit insist on its use as a point-of-care screening tool for SCD especially where there is a lack of laboratory facilities as well as in hospital-based set-up requiring immediate diagnosis and management of SCD. However, for further confirmation, the samples should be analyzed with other gold standard techniques like HPLC.

## Introduction

Sickle cell disease (SCD) is one of the most common monogenic inherited diseases in the world with an estimation of 111.91 per 1,00,000 live births across the globe [1]. With

**Funding:** This study was funded by Department of Health Research, Ministry of Health and Family Welfare, Government of India (Letter No. V.25011/570(i)/2010-HR Dt.07.01.2014) for the establishment of a Multi-Disciplinary Research Unit (MRU) at Maharaja Krishna Chandra Gajapati (M.K.C.G.) Medical College, Berhampur, Odisha, India. The funder has no role in study design, data collection and analysis, decision to publish, or preparation of manuscript.

**Competing interests:** The authors have declared that no competing interests exist.

significant morbidity and mortality, SCD is also a major health burden in India contributing about 15% of SCD newborns of the world as well as ranked as the second worst affected country for SCD globally [2, 3]. In India, the prevalence of SCD is mainly specific to certain populations residing in various eastern, central and western regions as well as predominantly prevalent in socio-economically disadvantaged ethnic groups [4, 5]. Overall, the prevalence of the sickle cell trait in India varies from 1–40% in various communities [5].

Sickle cell disease is defined as the presence of sickle hemoglobin (HbS: *HBB*; c.20T>A, p. Glu6Val) that polymerizes under low oxygen conditions leading to the formation of a crescent or sickle shaped red blood cells [6]. The inheritance of Hb S occurs either in the heterozygous condition (sickle cell trait/sickle cell carrier/AS) or in homozygous condition (homozygous sickle cell anemia/SCA/SS) or in double heterozygous conditions with other hemoglobin variants. The cases with SCD are characterized by asymptomatic to severe clinical manifestations and the most clinical presentations are episodes of painful events, anemia requiring blood transfusion, acute chest syndrome, jaundice, stroke etc [7, 8].

At present many methods are there for the screening of SCD including the sickling slide test, solubility test, Hb electrophoresis, Isoelectric focusing, HPLC and genetic testing [9–11]. Most of these methods can only be used in laboratories requiring a supply of electricity, trained manpower as well as the maintenance of equipment which is very difficult in resource-poor setting especially when carried out in tribal-dominated areas. For the early diagnosis as well as management of SCD, now the world is focused on point-of-care test [12]. A number of point-of-care rapid diagnostics test (RDT) kits are available for the diagnosis of SCD including HemoTypeSC, SickleSCAN, SickleDex, solubility test etc. with significant sensitivity and specificity. The use of RDT kits for the screening of SCD will be helpful for the early identification of SCD patients and the subsequent provision of comprehensive management to reduce the morbidity and mortality rate related to SCD. Further, the use of RDT kits does not require any sophisticated laboratory set-up. In this regard, the present study was undertaken for the validation as well as the usefulness of another point-of-care RDT kit (SICKLECHECK^TM) for the screening of SCD.

## Materials and methods

This was a validation, diagnostic accuracy study of the SICKLECHECK^TM RDT kit conducted among the patients advised for the screening of SCD. All the patients (more than 6 months of age) recruited during June 2023 to August 2023 for the screening of SCD were considered for this study. In an earlier hospital-based study carried out in this Medical College, the SCD cases was reported in around 50% of cases [13]. Considering the hypothesized sensitivity of 0.95, SCD prevalence of 0.50 and absolute precision of 0.05, the minimum number of SCD cases (sample size) required for this study was 146 [14]. However, a total sample size was fixed to 400 for the recruitment of normal individuals (Hb AA) for analysis.

After obtaining written informed consent from the patients (from the parents/guardian in case of a minor), 1–2 mL of venous blood samples were collected in an EDTA vial for laboratory investigations. All the collected blood samples were analyzed for both HPLC (Variant-II, Bio-Rad Laboratories, CA, USA) using β thalassemia short program and a rapid diagnostic test kit (SICKLECHECK^TM, Zephyr Biomedicals- A division of Tulip Diagnostics (P) Ltd., Goa, India) for the screening of SCD simultaneously. This study was approved by the Institutional Ethical Committee of M.K.C.G Medical College, Berhampur, Odisha (Ref No1298/Chairman-IEC, M.K.C.G.Medical College, Brahmapur-4, Date 05. 04.2023)

SICKLECHECK^TM RDT kit is a point-of-care, rapid, qualitative, immuno-chromatographic assay for the simultaneous detection of Hb S and Hb A in human blood samples.

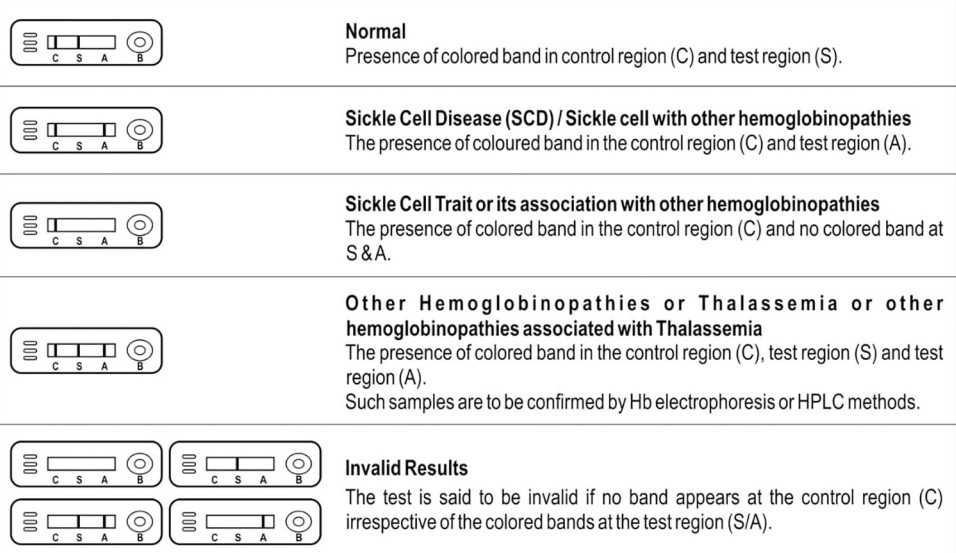

**Fig 1. Interpretation of results of SICKLECHECK^TM rapid diagnostic kit for the diagnosis of sickle cell diseases.**

It is based on the principle of agglutination of antibodies/antisera with respective antigens in a competitive immuno-chromatography format along with the use of nanogold particles as agglutination revealing agents. The designed test kit pad is impregnated with two components that is, monoclonal antibody for HbS conjugated to colloidal gold and monoclonal antibody for Hb A conjugated to colloidal gold. As the test sample flows through the membrane assembly of the kit, the highly specific monoclonal antibody for Hb S and Hb A conjugated with colloidal gold make complexes with the respective Hb S and Hb A antigens present in the test sample and travels on the membrane due to capillary action. The complex travels on the membrane to the test region (S and A mark in the kit) where it is not captured by Hb S and Hb A coated on the membrane, thus forming no bands. The absence of colour bands at the test regions (S and A) indicates the presence of respective HbS and HbA antigens in the test samples and vice versa. The unbound colloidal gold conjugates move further on the membrane and are immobilized by the agglutinating sera for goat anti-mouse IgG coated on the membrane at the control (C mark in the kit). This control band acts as a procedural control and serves to validate the test results. The detailed interpretation of results generated from the-SICKLECHECK^TM RDT kit is shown in Fig 1.

This SICKLECHECK^TM RDT kit contains membrane test assembly, plastic specimen transfer devices with a 10 μL mark, assay buffer vials and sterile lancets. The test can be performed either in venous blood or by finger prick. In both case, 10 μL of blood samples was mixed with supplied assay buffers by gently shaking the vial sideways. Two drops of the specimen were applied into the specimen port (B) of the membrane device and after 10–15 minutes the result was interpreted by observing the clear bands in the C, S and A regions respectively.

## Limit of detection

To analyze the minimum Hb S % that can be detected by SICKLECHECK^TM RDT kit, a serial dilution method was applied. For this, 5 different blood samples with 'AS' pattern and Hb S % of 20–30% in HPLC were considered and diluted with the normal ('AA' pattern) samples. These samples were diluted in 5:5, 4:6, 3:7, 2:8, 1:9 and 0.5:9.5 ration with the normal samples

and these diluted samples were simultaneously analyzed both in HPLC and SICKLE-CHECK<sup>TM</sup> RDT kit.

## Data analysis

The characteristics of the study participants were described using descriptive statistics. The generated results of both test were analyzed by respective instruction manuals and categorized the cases accordingly. The diagnostic accuracy of the SICKLECHECK<sup>TM</sup> RDT kit was compared with HPLC results as the gold standard by analyzing sensitivity, specificity, positive predictive value (PPV) and negative predictive value (NPV) using 2023 MedCalc Software Ltd. (http://www.medcalc.org/calc/diagnostic_test.php).

First, the diagnostic accuracy of the SICKLECHECK<sup>TM</sup> RDT kit was performed by analyzing the presence and absence of Hb S in the samples as follows

1. For HPLC

   - Presence of HbS: Cases with 'AS', 'SS', sickle cell and β thalassemia (HbS- β thalassemia), and SS with a history of blood transfusion.

   - Absence of HbS: Cases with 'AA', 'β thalassemia trait', 'β thalassemia major', 'hemoglobin E trait', 'hemoglobin Lepore trait', and 'trait for hereditary persistence of fetal hemoglobin'.

2. For SICKLECHECK<sup>TM</sup> RDT kit

   - Presence of Hb S: Cases with 'AS' and 'SS' pattern

   - Absence of Hb S: Cases with 'AA' and 'Others'(other hemoglobin variants).

Secondly, as theSICKLECHECK<sup>TM</sup> RDT kit is based on the screening for the SCD only, cases with 'AA', 'AS' and 'SS' patterns detected in HPLC were only considered for the diagnostic accuracy of the SICKLECHECK<sup>TM</sup> RDT kit.

## Results

During the study period, a total of 400 unknown samples were included in this study. The median age of the study participants was 19 years with a range from 7 months to 70 years. There were 204 females and 196 males. For the diagnosis of SCD in all the recruited cases both HPLC (Variant-II) and SICKLECHECK<sup>TM</sup> RDT were carried out simultaneously. In SICKLE-CHECK<sup>TM</sup> RDT, a total of 228, 110, 56 and 6 cases were detected as 'AA', 'AS', 'SS' and 'Others' patterns respectively. On comparison of both the investigations, various hemoglobin disorders including normal (AA), β thalassemia trait, hemoglobin E trait, hemoglobin Lepore trait and trait for hereditary persistence of fetal hemoglobin (high HbF %) diagnosed in HPLC were resulted with 'AA' pattern in SICKLECHECK<sup>TM</sup> RDT kit. Similarly, the 'AS' pattern and trait for hereditary persistence of fetal hemoglobin (high HbF %) diagnosed in HPLC resulted in the 'AS' pattern in the SICKLECHECK<sup>TM</sup> RDT kit. Cases with HbS-β thalassemia, 'SS' with a history of blood transfusion and 'SS' in HPLC resulted in 'SS' pattern in the SICKLE-CHECK<sup>TM</sup> RDT kit. All the β thalassemia major cases in HPLC were detected as 'Others' pattern in the SICKLECHECK<sup>TM</sup> RDT kit. The detailed comparative analysis of both HPLC and SICKLECHECK<sup>TM</sup> RDT kit is depicted in Table 1.

As per HPLC finding, Hb S was present in 164 cases and absent in rest 236 cases. Out of 164 cases with Hb S in HPLC, 163 cases had Hb S and a lone case had no Hb S in the SICKLE-CHECK<sup>TM</sup> RDT kit. Similarly, out of 236 cases with no Hb S in HPLC, 233 cases had no Hb S and 3 cases had Hb S in the SICKLECHECK<sup>TM</sup> RDT kit. With references to HPLC findings,

**Table 1. Comparison of results from HPLC (Variant II) and SICKLECHECK<sup>TM</sup> RDT kit (n = 400).**

| SICKLECHECK<sup>TM</sup> RDT kit | HPLC (Variant-II) |
|---|---|
| AA (n = 228) | AA (n = 200) |
| | AS (n = 1) |
| | β thalassemia trait (n = 20) |
| | Hemoglobin E trait (n = 1) |
| | HemoglobinLepore trait (n = 1) |
| | Trait for hereditary persistence of fetalhemoglobin (n = 5) |
| AS (n = 110) | AA (n = 2) |
| | AS (n = 106) |
| | Trait for hereditary persistence of fetalhemoglobin (n = 1) |
| | SS (n = 1) |
| SS (n = 56) | SS (n = 47) |
| | S-β thalassemias (n = 5) |
| | SS with history of blood transfusion (n = 4) |
| Others (n = 6) | AA (n = 2) |
| | β thalassemia major (n = 3) |
| | Trait for hereditary persistence of fetalhemoglobin (n = 1) |

SICKLECHECK<sup>TM</sup> RDT kit results showed a sensitivity of 99.39% and a specificity of 98.73% to detect the presence of Hb S in the samples (Fig 2).

As the SICKLECHECK<sup>TM</sup> RDT kit is designed for the diagnosis of SCD only, we have calculated the sensitivity and specificity from 359 cases including 204 cases of the 'AA' pattern, 107 cases of the 'AS' pattern and 48 cases of the 'SS' pattern resulted in HPLC. The detailed distribution of 359 cases is shown in Table 2. The sensitivity and specificity of the SICKLE-CHECK<sup>TM</sup> RDT kit were analyzed separately for the 'AA', 'AS' and 'SS' pattern. For the detection of the 'AS' pattern, the SICKLECHECK<sup>TM</sup> RDT kit has shown a sensitivity of 99.07% and a specificity of 98.81%. For the detection of the 'SS' pattern, the SICKLECHECK<sup>TM</sup> RDT kit has shown a sensitivity of 97.92% and a specificity of 100.0%. For the detection of the 'AA' pattern, the SICKLECHECK<sup>TM</sup> RDT kit has shown a sensitivity of 98.04% and a specificity of 99.35%. The sensitivity, specificity, PPV and NPV of theSICKLECHECK<sup>TM</sup> RDT kit for the detection of sickle cell disease are depicted in Table 3.

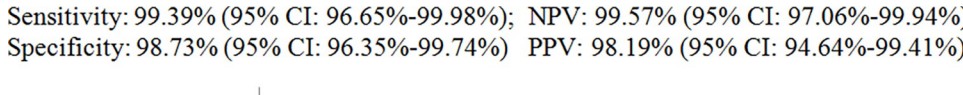

Sensitivity: 99.39% (95% CI: 96.65%-99.98%);   NPV: 99.57% (95% CI: 97.06%-99.94%)
Specificity: 98.73% (95% CI: 96.35%-99.74%)   PPV: 98.19% (95% CI: 94.64%-99.41%)

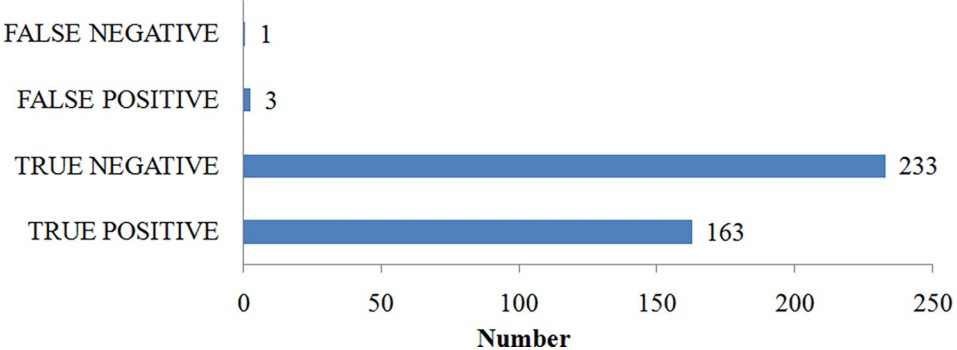

**Fig 2. Sensitivity and specificity of SICKLECHECK<sup>TM</sup> RDT kit for the presence of sickle hemoglobin.**

**Table 2. Distribution of cases with respect to HPLC (Variant-II) and SICKLECHECK[TM] RDT kit results (n = 359).**

| Sickle Cell Diseases | | SICKLECHECK[TM] RDT kit | | | | Total |
|---|---|---|---|---|---|---|
| | | 'AA' pattern | 'AS' pattern | 'SS' pattern | 'Others' pattern | |
| HPLC Results | AA | 200 | 2 | 0 | 2 | 204 |
| | | (98.04%) | (0.98%) | | (0.98%) | |
| | AS | 1 | 106 | 0 | 0 | 107 |
| | | (0.94%) | (99.36%) | | | |
| | SS | 0 | 1 | 47 | 0 | 48 |
| | | | (2.08%) | (97.92%) | | |
| Total | | 201 | 109 | 47 | 2 | 359 |

**Table 3. SICKLECHECK[TM] RDT kit sensitivity, specificity, positive predictive value and negative predictive value in comparison to HPLC findings for the detection of sickle cell diseases.**

| | SICKLECHECK[TM] RDT kit | | | | | |
|---|---|---|---|---|---|---|
| | 'AA' pattern | | 'AS' pattern | | 'SS' pattern | |
| | Value | 95% CI | Value | 95% CI | Value | 95% CI |
| Sensitivity | 98.04 | 95.06–99.46 | 99.07 | 94.90–99.98 | 97.92 | 88.93–99.95 |
| Specificity | 99.35 | 96.46–99.98 | 98.81 | 96.56–99.75 | 100.00 | 98.82–100.00 |
| Positive predictive value | 99.50 | 97.26–99.99 | 97.25 | 91.98–99.09 | 100.00 | 92.45–100.0 |
| Negative predictive value | 97.47 | 93.65–99.31 | 99.60 | 97.25–99.94 | 99.68 | 97.81–99.95 |

**Table 4. Cases with conflicting results (n = 6).**

| Case No. | Results of SICKLECHECK[TM] RDT kit | Results of HPLC | Hemoglobin variants from HPLC (%) | | | | Total hemoglobin level (mg/dL) |
|---|---|---|---|---|---|---|---|
| | | | HbA0 | HbA2 | HbF | HbS | |
| 1 | AS | AA | 84.5 | 2.6 | 0.6 | 0 | 11.3 |
| 2 | AS | AA | 80.1 | 2.7 | 0.4 | 0 | 7.7 |
| 3 | AS | SS | 6.5 | 2.7 | 14.3 | 73.3 | 7.3 |
| 4 | Other | AA | 88.1 | 3.2 | 0.9 | 0 | 11.9 |
| 5 | AA | AS | 67.4 | 3.8 | 4.1 | 17.0 | 11.7 |
| 6 | Other | AA | 84.2 | 2.7 | 1.8 | 0 | 10.9 |

In the comparison of both the HPLC and SICKLECHECK[TM] RDT kit in 359 cases, 6 cases showed conflicting results with respect to each test. As the SICKLECHECK[TM] RDT kit has certain limitations for lower hemoglobin level, high fetal hemoglobin level, percentage of both Hb S and Hb A in the samples etc., the detailed hemoglobin level along with HPLC finding of 6 contradictory cases is shown in Table 4. None of these cases had significant findings that can be correlated with the limitations of the SICKLECHECK[TM] RDT kit.

The limit of detection for SICKLECHECK[TM] RDT kit for the detection of Hb S in the blood samples were analyzed with a serial dilution and observed that samples with less than 3% of Hb S had an appearance of 'S' line indicating the absence of Hb S in the samples. So a limit of detection of 3% of Hb S in the samples may be considred for the detection of Hb S by SICKLECHECK[TM] RDT kit.

## Discussion

During this validation study, the result of SICKLECHECK[TM] RDT kit was compared with the HPLC as a gold standard technique for the screening of SCD. In comparison, the

SICKLECHECK<sup>TM</sup> RDT kit was found to have high sensitivity and specificity for the screening of SCD. It showed high sensitivity and specificity of more than 98% in both when compared to either the presence or absence of Hb S in the samples; or the genotypes (AA, AS, SS) of the individuals respectively. These observations insist on the use of the SICKLECHECK<sup>TM</sup> RDT kit as a screening tool for SCD.

SICKLECHECK<sup>TM</sup> RDT kit can be used in remote areas without requiring sophisticated equipment, electricity, laboratory etc. Only minimum training is required for interpretation of results. Secondly, the results can be read within 15 minutes of the blood collection compared to 24–48 hours for HPLC (from sample collection to reporting) affirming its use in many hospital set-ups necessitating immediate diagnosis of the SCD for proper management and treatment. Further, the estimated price per kit is 1.3 USD, which is comparatively low than HPLC (~10 USD) and is also comparable to other point-of-care tests. In the state of Odisha with high prevalence of SCD as well as inhabitants of many tribal communities especially residing in remote areas with minimal or no health facilities, the SICKLECHECK<sup>TM</sup> RDT kit will be useful as a point-of-caretest for SCD. The use of point-of-care RDT for the screening of SCD will lead to early identification of disease and subsequent provision of comprehensive care. The comprehensive care management of SCD patients especially during childhood has been associated with a significant reduction in morbidity and mortality [15].

During the study, a number of other Hb variants were also reported and some of them were also co-inherited with Hb S. Interestingly, all the double heterozygous states of Hb S and β thalassemia diagnosed by HPLC were reported as 'SS' patterns in SICKLECHECK<sup>TM</sup> RDT kit. Again, all four 'SS' cases with a history of blood transfusion (HbA0 level was 10–16% in HPLC) were also reported as 'SS' pattern in the SICKLECHECK<sup>TM</sup> RDT kit. Both of these observations also affirm its use in other variants of SCD including double heterozygous state of Hb S and β thalassemia which is the second most prevalent severe phenotype of SCD after 'SS' genotype worldwide including India [16]. However, its pattern in another double heterozygous state with Hb S needs to be investigated further. Though, all the 'SS' cases with blood transfusion were resulted with 'SS' pattern, we recommend to test after 3 months of blood transfusion for any false negative results.

This SICKLECHECK<sup>TM</sup> RDT kit has also been designed for the diagnosis of other Hb variants. Except few, the majority of Hb variants including β thalassemia trait, hemoglobin E trait, hemoglobin Lepore trait and trait for hereditary persistence of fetal hemoglobin (high HbF %) diagnosed by HPLC in this study were resulted with 'AA' pattern in SICKLECHECK<sup>TM</sup> RDT kit. All three β thalassemia major cases in HPLC were detected as 'Others' patterns in the SICKLECHECK<sup>TM</sup> RDT kit. Thus, the diagnosis of other Hb variants (other than Hb S) by SICKLECHECK<sup>TM</sup> RDT kit needs to be revised requiring further upgradation of membrane biology. In such a finding, the samples should be analyzed by HPLC for further confirmation.

On estimating the limit of detection ofSICKLECHECK<sup>TM</sup> RDT kit, the serial dilution showed that the SICKLECHECK<sup>TM</sup> RDT kit can detect a minimum of 3% of Hb S in the samples. With a 3% of limit of detection in the present study, the SICKLECHECK<sup>TM</sup> RDT kit may be useful for the screening of SCD in newborn. However, further necessary evaluation is required for its use in newborns screening on SCD.

Two mostly validated and recommended RDT kit for the screening of SCD were SickleSCAN and HemoTypeSC. SickleSCAN assay employs the sandwich format chromatographic immunoassay using polyclonal antibodies specific to Hb A, Hb S and Hb C; while HemoTypeSC assay are based on the competitive lateral flow immuno-chromatographic assay using specific antibodies against Hb A, Hb S and Hb C [17, 18]. Both the RDT kits have been validated and evaluated in many studies for its wider use. Here, SICKLECHECK<sup>TM</sup> RDT kit used the same principle like HemoTypeSC, but there is a feasibility of detection of 'Other' variants

instead of Hb C alone. As Hb C variant is a rare Hb disorder in India with very few case reports, the detection of 'Other' variants is important as designed in SICKLECHECK[TM]RDT kit in country like India where other Hb variants are highly prevalent including β thalassemia, Hb D[Punjab], Hb E etc.

The use of competitive immuno-chromatographic assay in the SICKLECHECK[TM] RDT kit enhances the specificity of the test results by avoiding the possible interferences from other proteins in the samples as well as can allow the detection of a wide range of concentrations of target molecules (Hb A and Hb S). It may also exhibit lower background signals in the absence of the target Hb, which can contribute to improve the signal-to-noise ratios resulting better sensitivity in certain situations. Further this competitive immunoassaymay be a better option from the sandwich type immunoassay which are usually susceptible to the prozone effect (hook effect), in which a high concentrations of a particular Hb may leading to a saturation of the binding sites on the labelled conjugate and capture reagent on the test line, as a result, the excess Hb competes less effectively with the labelled conjugate, causing a decrease in the signal at the test line yielding a false negative results [19–21].

There are certain limitations of the study; (1) it does not quantify the amount of Hb S and Hb A in the samples; (2) The presence of heterophilic substances in the samples and lipemic or icteric samples may give incorrect results though no such type of samples was analyzed in this study. (3) The presence of a control line only means that the migration of the specimen is occurred. It does not serve as confirmation of the addition of more test specimens.

In conclusion, the SICKLECHECK[TM] RDT kit can be used as a point-of-care test for the screening of SCD only. The readability of results in the SICKLECHECK[TM] RDT kit is easy and clear. A clear and dark band in the test membrane should be considered for diagnosis; while the appearance of any faint or light bands should go for further confirmation with other gold standard techniques like HPLC.

## Supporting information

**S1 Dataset.**
(XLSX)

## Acknowledgments

The authors acknowledge laboratory technician, Mrs RadhaRani Dora and Mr. Himansu Sekhar Sahu for their support in collection and processing of blood samples during the investigations. The authors also acknowledge to Mr. Satyanarayana Mohanty, Data Entry Operator, Multi-Disciplinary Research Unit (MRU) for data processing. For use of the SICKLE-CHECK[TM] Kit information or photographs, necessary permission has been obtained from the Zephyr Biomedicals- A division of Tulip Diagnostics (P) Ltd., Goa, India.

## Author Contributions

**Conceptualization:** Prasanta Purohit, Samira Kumar Behera.

**Data curation:** Prasanta Purohit.

**Investigation:** Prasanta Purohit, Chinmayee Parida, Tapan Kumar Martha, Snehal Bholo.

**Methodology:** Prasanta Purohit, Samira Kumar Behera.

**Project administration:** Samira Kumar Behera.

**Supervision:** Samira Kumar Behera.

**Writing – original draft:** Prasanta Purohit, Aparupa Naik.

**Writing – review & editing:** Prasanta Purohit, Samira Kumar Behera.

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
