## [Decision Letter · Decision Letter 0]

1 May 2024

PONE-D-24-04629Evaluation of a point-of-care rapid diagnostic test kit (SICKLECHECKTM) for screening of sickle cell disordersPLOS ONE

Dear Dr. Behera,

Thank you for submitting your manuscript to PLOS ONE. After careful consideration, we feel that it has merit but does not fully meet PLOS ONE’s publication criteria as it currently stands. Therefore, we invite you to submit a revised version of the manuscript that addresses the points raised during the review process.

We look forward to receiving your revised manuscript.

Kind regards,

Ibrahim Sebutu Bello, MBBS, MPH, MD, FMCGP

Academic Editor

PLOS ONE

Journal Requirements:

2. Thank you for stating the following financial disclosure: "This study was funded by Department of Health Research, Ministry of Health and Family Welfare, Government of India (Letter No. V.25011/570(i)/2010-HR Dt.07.01.2014) for the establishment of a Multi-Disciplinary Research Unit (MRU) at Maharaja Krishna Chandra Gajapati Medical College, Berhampur, Odisha, India." 

Reviewers' comments:

Reviewer's Responses to Questions

**Comments to the Author**

1. Is the manuscript technically sound, and do the data support the conclusions?

Reviewer #1: Partly

Reviewer #2: Yes

2. Has the statistical analysis been performed appropriately and rigorously? 

Reviewer #1: No

Reviewer #2: Yes

3. Have the authors made all data underlying the findings in their manuscript fully available?

Reviewer #1: Yes

Reviewer #2: No

4. Is the manuscript presented in an intelligible fashion and written in standard English?

Reviewer #1: Yes

Reviewer #2: Yes

5. Review Comments to the Author

Reviewer #1: Introduction may be shortened. Rationale for conducting this validation study should be highlighted.

Sample size needs to be recalculated.

calculation of sensitivity and specificity for differentiating SS with AS should be rechecked.

In tables percentage of AA, AS and SS should be given.

approximate cost per test should be given.

Limit of detection of HbS and HbA should be tested.

How SICKLECHECK is comparable to other RDTs available in the market should be discussed in detail.

Reviewer #2: The manuscript describes an Evaluation of a point-of-care rapid diagnostic test kit (SICKLECHECKTM) for screening of sickle cell disorders, comparing it with HPLC for diagnosis. The manuscript is prospective and descriptive comparing the new screening test against the commonly used gold standard. It is well written; however, some points need to be addressed.

General suggestion:

1. Use full forms of all abbreviations at first use- both in tables and text.

Abstract:

1. .

Introduction

1. Instead of mentioning various states and disadvantaged groups- keep it general as Central India/ tribal belt, for a general audience of the journal.

2. General discussion on SCD can be omitted.

Methods

1. Blinding of the hospital name is advisable when describing the hospital setting.

2. Please mention.

a. Inclusion and exclusion criteria in more detail.

b. Were all consecutive patients enrolled, age group?

c. Any specific criteria for screening or are all patients considered for testing?

d. What sample was used in the current study- venous blood or finger prick?

e. Any time gap recommended from the last blood transfusion before testing the individuals?

3. Whether the kit results are influenced by blood transfusions?

4. Did the authors perform genetic testing for discordant results for confirmation of diagnosis? This will be beneficial for overall study findings.

5. Funding agency details can be omitted from the methods section.

Results

1. Repetition of results in text and tables can be minimized.

2. It will be helpful, if authors can provide the median time to diagnosis of SCD, in their settings or previously published data in their region.

Discussion

1. Comparative analysis with other POC test kits can be added to text (in short) and detail in a table.

Tables

1. Tables 1 and 2 appear to convey the same information. Only Table 2 be retained.

2. For table 5, it will be better if the final diagnosis of the patients is also listed, including the reason for low Hb.

Figure:

1. Figures 1 and 3 convey similar findings, either of them can be omitted.

2. For a scientific journal, utility of figure 2 is also minimal- can be omitted.

3. Figure 4 and table 4 are giving same information. Figure 4 can be omitted.

6. PLOS authors have the option to publish the peer review history of their article (what does this mean?). If published, this will include your full peer review and any attached files.

Reviewer #1: No

Reviewer #2: **Yes: **Dr Mukul Aggarwal

---

## [Author Response · Author response to Decision Letter 0]

5 Jun 2024

Response to Reviewers 

PONE-D-24-04629

Evaluation of a point-of-care rapid diagnostic test kit (SICKLECHECKTM) for screening of sickle cell disorders

Comments: If you would like to make changes to your financial disclosure, please include your updated statement in the cover letter. If applicable, we recommend that you deposit your laboratory protocols in protocols.io to enhance the reproducibility of your results. Protocols.io assigns your protocol its own identifier (DOI) so that it can be cited independently in the future. For instructions see: https://journals.plos.org/plosone/s/submission-guidelines#loc-laboratory-protocols. Additionally, PLOS ONE offers an option for publishing peer-reviewed Lab Protocol articles, which describe protocols hosted on protocols.io. Read more information on sharing protocols at https://plos.org/protocols?utm_medium=editorial-email&utm_source=authorletters&utm_campaign=protocols.

Response: Thank you for suggestions, but we want to publish it along with the original research article. 

Journal Requirements:

Comments: 1. Please ensure that your manuscript meets PLOS ONE's style requirements, including those for file naming. The PLOS ONE style templates can be found at 

Response: Necessary modifications have been incorporated in the revised manuscript

Comments: 

2. Thank you for stating the following financial disclosure: "This study was funded by Department of Health Research, Ministry of Health and Family Welfare, Government of India (Letter No. V.25011/570(i)/2010-HR Dt.07.01.2014) for the establishment of a Multi-Disciplinary Research Unit (MRU) at Maharaja Krishna Chandra Gajapati Medical College, Berhampur, Odisha, India." Please state what role the funders took in the study. If the funders had no role, please state: ""The funders had no role in study design, data collection and analysis, decision to publish, or preparation of the manuscript."" If this statement is not correct you must amend it as needed. Please include this amended Role of Funder statement in your cover letter; we will change the online submission form on your behalf.

Response: Modified as per the suggestions

Comments: 

Response: Data Availability statement has been modified accordingly. Data will be available as supplementary files

Reviewers' comments:

Comments to the Author

1. Is the manuscript technically sound, and do the data support the conclusions?

Reviewer #1: Partly

Reviewer #2: Yes

Response: Many changes have been incorporated in the revised manuscript

Comments: 2. Has the statistical analysis been performed appropriately and rigorously?

Reviewer #1: No

Reviewer #2: Yes

Response: Necessary modifications have been incorporated in the revised manuscript

Comments: 3. Have the authors made all data underlying the findings in their manuscript fully available? The PLOS Data policy requires authors to make all data underlying the findings described in their manuscript fully available without restriction, with rare exception (please refer to the Data Availability Statement in the manuscript PDF file). The data should be provided as part of the manuscript or its supporting information, or deposited to a public repository. For example, in addition to summary statistics, the data points behind means, medians and variance measures should be available. If there are restrictions on publicly sharing data—e.g. participant privacy or use of data from a third party—those must be specified.

Reviewer #1: Yes

Reviewer #2: No

Response: Data will be available as supplementary file (Data Availability statement has also been modified accordingly)

Comments: 4. Is the manuscript presented in an intelligible fashion and written in standard English? PLOS ONE does not copyedit accepted manuscripts, so the language in submitted articles must be clear, correct, and unambiguous. Any typographical or grammatical errors should be corrected at revision, so please note any specific errors here.

Reviewer #1: Yes

Reviewer #2: Yes

Response: As per various suggestions from journal Editorials and reviewers, required changes have been incorporated.

Comments: 5. Review Comments to the Author

Reviewer #1: Introduction may be shortened. Rationale for conducting this validation study should be highlighted.

Response: Necessary changes have been incorporated in the revised manuscript as suggested

Comments: Sample size needs to be recalculated.

Response: Sample size has been recalculated as per the suggestion. 

Comments: Calculation of sensitivity and specificity for differentiating SS with AS should be rechecked.

Response: As per the suggestions, we have re-checked and necessary modifications have been incorporated (in revised table-3)

Comments: In tables percentage of AA, AS and SS should be given.

Response: Revised 

Comments: approximate cost per test should be given.

Response: Mentioned in the discussion part.

Comments: Limit of detection of HbS and HbA should be tested.

Response: Limit of detection for SICKLECHECK has been analyzed and incorporated in the revised manuscript both in methods, results and discussion parts

Comments: How SICKLECHECK is comparable to other RDTs available in the market should be discussed in detail.

Response: A comparative analysis of other RDT kits have been discussed in details in the revised manuscript 

Comments 

Reviewer #2: The manuscript describes an Evaluation of a point-of-care rapid diagnostic test kit (SICKLECHECKTM) for screening of sickle cell disorders, comparing it with HPLC for diagnosis. The manuscript is prospective and descriptive comparing the new screening test against the commonly used gold standard. It is well written; however, some points need to be addressed.

General suggestion:

Comments: 1. Use full forms of all abbreviations at first use- both in tables and text.

Response: Necessary modifications have been incorporated

Comments: Abstract:

1. .Introduction

1. Instead of mentioning various states and disadvantaged groups- keep it general as Central India/ tribal belt, for a general audience of the journal.

Response: Modified as per the suggestions

2. General discussion on SCD can be omitted.

Response: Modified as per the suggestions

Comments: Methods

1. Blinding of the hospital name is advisable when describing the hospital setting.

Response: Modified as per the suggestions

Comments:2. Please mention.

a. Inclusion and exclusion criteria in more detail.

Response: Modified as per the suggestions

Comments: b. Were all consecutive patients enrolled, age group?

Response: All the consecutive patients were enrolled for this study. Age range: more than 6 months of age. Already mentioned in the manuscript in different sections. 

Comments:c. Any specific criteria for screening or are all patients considered for testing?

Response: All patients were enrolled for this study. 

Comments:d. What sample was used in the current study- venous blood or finger prick?

Response: Venous blood. However both venous blood and finger prick can be used for testing (mentioned in the manuscript) 

Comments:e. Any time gap recommended from the last blood transfusion before testing the individuals?

Response: As all the ‘SS’ cases with blood transfusion were resulted with ‘SS’ pattern. No time gap is fixed for blood transfusion. But recommended to test after 3 months of blood transfusion for clear results. 

Comments: 3. Whether the kit results are influenced by blood transfusions?

Response: No. All the 4 ‘SS’ cases with blood transfusion were resulted with ‘SS’ pattern. However we have recommended to test after 3 months of blood transfusion for clear results. 

Comments: 4. Did the authors perform genetic testing for discordant results for confirmation of diagnosis? This will be beneficial for overall study findings.

Response: No, we have only compared with the HPLC (variant-II) results 

Comments: 5. Funding agency details can be omitted from the methods section.

Response: Modified as per the suggestions

Comments: Results: 1. Repetition of results in text and tables can be minimized.

Response: Modified as per the suggestions

Comments: 2. It will be helpful, if authors can provide the median time to diagnosis of SCD, in their settings or previously published data in their region.

Response: Modified as per the suggestions and mentioned in the discussion section.

Discussion

Comments: 1. Comparative analysis with other POC test kits can be added to text (in short) and detail in a table. 

Response: Modified as per the suggestions and mentioned in the discussion section.

Tables

Comments: 1. Tables 1 and 2 appear to convey the same information. Only Table 2 be retained.

Response: Modified as per the suggestions

Comments: 2. For table 5, it will be better if the final diagnosis of the patients is also listed, including the reason for low Hb. 

Response: As we have collected the samples from the testing units, the details of final diagnosis is unavailable. 

Figure:

Comments: 1. Figures 1 and 3 convey similar findings, either of them can be omitted.

Response: Modified as per the suggestions

Comments: 2. For a scientific journal, utility of figure 2 is also minimal- can be omitted.

Response: Omitted as per the suggestions

Comments: 3. Figure 4 and table 4 are giving same information. Figure 4 can be omitted.

Response: The information in Figure 4 (now figure 2) and table 4 (now table 3) are different. Figure 2 is for the sensitivity and specificity calculated for the presence and absence of Hb S (from 400 cases); while Table 3 is for genotypes pattern i.e. AA, AS and SS (359 cases). 

Comments: While revising your submission, please upload your figure files to the Preflight Analysis and Conversion Engine (PACE) digital diagnostic tool, https://pacev2.apexcovantage.com/. PACE helps ensure that figures meet PLOS requirements. To use PACE, you must first register as a user. Registration is free. Then, login and navigate to the UPLOAD tab, where you will find detailed instructions on how to use the tool. If you encounter any issues or have any questions when using PACE, please email PLOS at figures@plos.org. Please note that Supporting Information files do not need this step.

Response: Modified accordingly as per the suggestions

Comments: (in track changes of pdf file) 

In the above para it is mentioned that HbS was present in 164 cases in HPLC and 163 cases in Sicklecheck. Now authors calculate the sensitivity using 155 samples only. It is contradictory.

Response: Initially the analyses were made for the presence and absence of Hb S in all the recruited 400 cases. Further, as the SICKLECHECKTM RDT kit is designed for the diagnosis of SCD only, we have calculated the sensitivity and specificity from 359 cases including 204 cases of the ‘AA’ pattern, 107 cases of the ‘AS’ pattern and 48 cases of the ‘SS’ pattern resulted in HPLC. Both the analyses have been mentioned in the methods section.

---

## [Decision Letter · Decision Letter 1]

16 Jul 2024

PONE-D-24-04629R1Evaluation of a point-of-care rapid diagnostic test kit (SICKLECHECKTM) for screening of sickle cell diseasesPLOS ONE

Dear Dr. Behera,

Thank you for submitting your manuscript to PLOS ONE. After careful consideration, we feel that it has merit but does not fully meet PLOS ONE’s publication criteria as it currently stands. Therefore, we invite you to submit a revised version of the manuscript that addresses the points raised during the review process.

We look forward to receiving your revised manuscript.

Kind regards,

Ibrahim Sebutu Bello, MBBS, MPH, MD, FMCGP

Academic Editor

PLOS ONE

Journal Requirements:

Reviewers' comments:

Reviewer's Responses to Questions

**Comments to the Author**

1. If the authors have adequately addressed your comments raised in a previous round of review and you feel that this manuscript is now acceptable for publication, you may indicate that here to bypass the “Comments to the Author” section, enter your conflict of interest statement in the “Confidential to Editor” section, and submit your "Accept" recommendation.

Reviewer #1: All comments have been addressed

Reviewer #2: All comments have been addressed

2. Is the manuscript technically sound, and do the data support the conclusions?

Reviewer #1: Yes

Reviewer #2: Yes

3. Has the statistical analysis been performed appropriately and rigorously? 

Reviewer #1: Yes

Reviewer #2: Yes

4. Have the authors made all data underlying the findings in their manuscript fully available?

Reviewer #1: Yes

Reviewer #2: Yes

5. Is the manuscript presented in an intelligible fashion and written in standard English?

Reviewer #1: Yes

Reviewer #2: Yes

6. Review Comments to the Author

Reviewer #1: In discussion, it is written that HPLC took 24-48 hours for results which is not correct. HPLC required only 6.5 minutes per sample.

"the double heterozygous states of HbS and β thalassemia diagnosed by HPLC" HPLC cannot differentiate homozygous sickle cell and S-Beta thalassemia.

Reviewer #2: Authors have addressed most of the reviewer's comments. I feel discussion and introduction section can be shortened further, minimize general information on the disease.

7. PLOS authors have the option to publish the peer review history of their article (what does this mean?). If published, this will include your full peer review and any attached files.

Reviewer #1: **Yes: **Dr Ravindra Kumar

Reviewer #2: **Yes: **Dr Mukul Aggarwal

---

## [Author Response · Author response to Decision Letter 1]

31 Jul 2024

Review Comments to the Author

Reviewer-1

Comments : In discussion, it is written that HPLC took 24-48 hours for results which is not correct. HPLC required only 6.5 minutes per sample.

Response: It was written from samples collections to result dispatched. Accordingly, the sentence has been corrected in the revised manuscript.

Comments: "the double heterozygous states of HbS and β thalassemia diagnosed by HPLC" HPLC cannot differentiate homozygous sickle cell and S-Beta thalassemia.

Response: We agree with the reviewer’s comment. The double heterozygous states of HbS and β thalassemia has been diagnosed on the basis of both HbA2 level as well as red cell pictures and parents study where required. 

Reviewer 2

Comments: Authors have addressed most of the reviewer's comments. I feel discussion and introduction section can be shortened further, minimize general information on the disease.

Response: Necessary corrections have been incorporated in the revised manuscript.

---

## [Decision Letter · Decision Letter 2]

6 Aug 2024

Evaluation of a point-of-care rapid diagnostic test kit (SICKLECHECKTM) for screening of sickle cell diseases

PONE-D-24-04629R2

Dear Dr. Behera,

We’re pleased to inform you that your manuscript has been judged scientifically suitable for publication and will be formally accepted for publication once it meets all outstanding technical requirements.

Kind regards,

Ibrahim Sebutu Bello, MBBS, MPH, MD, FMCGP

Academic Editor

PLOS ONE

Reviewers' comments:

Reviewer's Responses to Questions

**Comments to the Author**

1. If the authors have adequately addressed your comments raised in a previous round of review and you feel that this manuscript is now acceptable for publication, you may indicate that here to bypass the “Comments to the Author” section, enter your conflict of interest statement in the “Confidential to Editor” section, and submit your "Accept" recommendation.

Reviewer #1: All comments have been addressed

Reviewer #2: All comments have been addressed

2. Is the manuscript technically sound, and do the data support the conclusions?

Reviewer #1: Yes

Reviewer #2: Yes

3. Has the statistical analysis been performed appropriately and rigorously? 

Reviewer #1: Yes

Reviewer #2: Yes

4. Have the authors made all data underlying the findings in their manuscript fully available?

Reviewer #1: Yes

Reviewer #2: Yes

5. Is the manuscript presented in an intelligible fashion and written in standard English?

Reviewer #1: Yes

Reviewer #2: Yes

6. Review Comments to the Author

Reviewer #1: Authors have revised the manuscript. All my comments have been addressed. All references must be written as per journal's style.

Reviewer #2: The authors have addressed all necessary concerns from reviewers. Thanks

No further suggestions from my side.

7. PLOS authors have the option to publish the peer review history of their article (what does this mean?). If published, this will include your full peer review and any attached files.

Reviewer #1: **Yes: **Ravindra Kumar

Reviewer #2: **Yes: **Dr Mukul Aggarwal

---

## [Editor Report · Acceptance letter]

8 Aug 2024

PONE-D-24-04629R2 

PLOS ONE

Dear Dr. Behera, 

I'm pleased to inform you that your manuscript has been deemed suitable for publication in PLOS ONE. Congratulations! Your manuscript is now being handed over to our production team.

Kind regards, 

on behalf of

Dr. Ibrahim Sebutu Bello 

Academic Editor

PLOS ONE